# Cognitive Radio Strategy Combined with MODCOD Technique to Mitigate Interference on Low-Orbit Satellite Downlinks

**DOI:** 10.3390/s23167234

**Published:** 2023-08-17

**Authors:** Rodolfo Araujo, Luciano da Silva, Walter Santos, Marcelo Souza

**Affiliations:** 1Division of Space Electronics and Computing, National Institute for Space Research, São José dos Campos 12227-010, SP, Brazil; luciano.silva@inpe.br; 2Division of Small Satellites, National Institute for Space Research, São José dos Campos 12227-010, SP, Brazil; walter.abrahao@inpe.br; 3Space Engineering and Technology Course-ETE, National Institute for Space Research, São José dos Campos 12227-010, SP, Brazil; marcelosouzabra@gmail.com

**Keywords:** cognitive radio, downlink throughput, low-orbit satellite, MODCOD, spectral coexistence

## Abstract

The concept of cognitive radio (CR) as a tool to optimize the obstacle of spectral coexistence has promoted the development of shared satellite–terrestrial wireless networks. Nevertheless, in some applications like Earth Exploration Satellite Services, which demand high spectral efficiency (bps/Hz) for downlink transmissions, spectral coexistence amidst interferences from cellular Base Stations is still challenging. Our research aims to mitigate these interferences on low-orbit satellite downlinks carrying imaging data received from a ground station. In order to fulfill this, we present cognitive radio approaches to enhance spectrum exploitation and introduce the adaptive modulation and coding (MODCOD) technique to increase RF power and spectral efficiencies. Therefore, we propose a combined methodology using CR and adaptive MODCOD (ACM) techniques. Afterwards, we applied the solution by monitoring the signal to interference plus noise ratio and the MODCOD strategy. Finally, we provide a real in situ case study at the Cuiabá ground station located in Brazil’s central area, which receives images from an Earth observation satellite (EOS). In addition to demonstrating the strategy effectiveness in this scenario, we conducted a bench test emulating the interfering wireless communication system. In this sense, we demonstrated the proposed approach, successfully mitigating the harmful effects on the received EOS images.

## 1. Introduction

In recent years, the ever-increasing demand for wireless communication services has led to a significant scarcity of available radio spectrum [1]. The traditional approach of exclusive spectrum allocation to diverse services and applications has become inefficient and unsustainable [2]. Spectrum scarcity has motivated the progression of the cognitive radio (CR) communication concept, which comprises a variety of techniques allowing spectral coexistence between primary and secondary users [3].

In this article, we adapt/implement operational approaches to deal with the interferences. Innovative methods of spectral exploitation for high-throughput communication satellites combined with CR paradigm architecture will promote the solution to achieve the required high data rate and spectrum sharing goals in the coming years [3,4,5,6,7,8]. The entire concept of CR was proposed and detailed by Joseph Mitola III in his Ph.D. thesis in 2000 [9]. In [10], cognitive radio techniques are identified for implementing satellite communication solutions.

For the terrestrial-satellite context, widely used approaches [11,12,13] allow the sharing of the available spectrum between primary and secondary users (PUs and SUs). In the EOS (Earth observation satellite) downlink, we can improve the performance of high data rate transmitters to meet bandwidth requirements and efficiently utilize RF energy by employing modulation and coding (MODCOD) techniques in an ACM (adaptive MODCOD) system [14,15]. Simultaneous transmissions of primary and secondary users will be necessary in the same spectral band at the same time and location, for example, are described in [16]: a GEOstationary satellite as the SU with a Low Earth Orbit (LEO) satellite as the PU, or in [17,18]: a LEO satellite system as the SU.

Conversely, in contrast with what is presented in [13], where the satellite and the terrestrial system operate as the primary network and the cognitive network, respectively, our work focuses on the satellite communication system and its reception of remote-sensing Earth-observation images amidst interference in a scenario where both are primary users. Investigating the performance of satellite downlinks in the presence of interfering signals plus noise becomes a crucial obstacle to be addressed. In this sense, this paper aims at verifying and mitigating the harmful effects on the received images.

By the early 1970s, the leading Brazilian satellite earth station (ERG) was installed on a site far from the urban area of Cuiabá City, positioned in the geodetic center of South America. However, due to the region’s fast urbanization, the new Business and Administrative Center of Cuiabá became closer to the ERG, bringing together advanced microwave FS links and inducing unbearable spectral interference at specific points.

Earth Exploration Satellite Services (EESS) missions in Brazil monitor agriculture and the diverse biomes, including deforestation in the Amazon rainforest [19,20]. Additionally, these missions aim at deploying small and medium-sized satellites (PqSats), with payloads utilizing ultra-high resolution optical sensors or Synthetic Aperture Radars (SARs) that require/produce high data rates [21] for downlink transmissions, involving the sharing of their frequency spectrum with Fixed Service (FS) systems, which are a category of the Private Limited Service-SLP regulated by the Brazilian National Telecommunication Agency (ANATEL) [22], for example, cellular Base Stations (BS) at Cuiabá, Mato Grosso, Brazil. This scenario highlights the issues of spectral efficiency (in bps/Hz) and coexistence in frequency, time, location, and polarization.

Our research provides a solution to mitigate interference in the LEO satellite down-links of Earth exploration services through CR and adaptive MODCOD techniques (ACM) based on DVB-S2X. The primary approach is to manipulate the downlink transmission parameters in the Earth observation communication system (onboard TX and ground station RX) by using CR techniques in the underlay paradigm to remove degradation in the image caused by these interferences. This combined strategy allows spectral, temporal, and spatial coexistence, achieving the required throughput. The contribution of this work addresses the mitigation of harmful interference effects in coexisting systems by evaluating case studies of EOS with images from a WFI camera (wide field imager). We conducted bench tests to determine the no-failures reception threshold as a function of the signal-to-interference plus noise ratio.

The article is organized as follows: Section 2 presents the materials and methods for CR paradigm exploitation, MODCOD in ACM, and our combined solution for the EESS downlink coexistence scenario, along with the strategy to mitigate the harmful effects caused by interference in the desired signal. Section 3 presents the main empirical-analytical results for the chosen combined MODCOD/CR strategy in the case studies and the Lab emulation of the communication system. The results are discussed in Section 4. A brief conclusion of the work is provided in Section 5.

## 2. Materials and Methods

A brief overview of CR spectrum exploitation techniques and the interfering scenario is presented to calculate the link budget with real parameters.

### 2.1. Cognitive Radio Paradigms and Techniques

Cognitive radio system (CRS) is defined in [10] as follows:

“A CRS employs technology that allows the radio to comprehend its internal state, the operational and geographical environment, and the established regulations to dynamically and autonomously adjust its parameters and operational protocols based on the acquired knowledge to achieve predefined objectives.”

Considering that, we propose to use the concept of CR to address the challenge of spectrum scarcity in the development of shared satellite–terrestrial wireless networks. We can describe the operational capabilities available in proposed CRS as continuous cognitive cycles [3] (of a “control system”) in the following sequence (see Figure 1):Spectrum awareness (“Sensor”)—the first task for a CR is to be aware of its surrounding radio environment;Analysis and decision (“Controller”)—analyze the obtained information and make an intelligent decision on how to use the available resources effectively;Spectrum exploitation and adaptation (“Actuator”)—in other words, in any environmental conditions, the CR autonomously adapts its operational parameters, such as transmission power, operating frequency, modulation and coding schemes, antenna pattern, or polarization, to effectively exploit the available spectral opportunities.

In this sense, the spectrum awareness capability allows a CR to obtain information about dynamic spectral opportunities. In contrast, the spectrum exploitation capability assists the CR in efficiently exploiting the spectral availabilities.

Based on the level of knowledge of PU transmission signals, existing spectrum exploitation techniques can be classified into the interweave, underlay, and overlay paradigms. The underlay, which is employed in our practical context, can be defined as [23]:

Underlay—in this paradigm, simultaneous cognitive and non-cognitive transmissions are allowed in the same frequency band as long as the level of interference on the PU side remains acceptable at the defined threshold (see Figure 2). The maximum allowable interference level can be modeled by the concept of interference temperature [24], and it is regulated by local federal policy (e.g., FCC Spectrum Policy Task Force in [25]). Communication system information such as signal-to-interference plus noise ratio (SINR), signal and channel parameters, and DoAs (directions of arrival) are necessary to perform underlay techniques. The required SINR fluctuates according to the used MODCOD.

### 2.2. Modulation and Coding (MODCOD) Concept and ACM Technique Based on DVB-S2X

The concept of MODCOD is based on the rational application of highly efficient and enhanced modulation for non-linearities, combined with efficient channel error correction codes [26]. Well-recognized techniques for high data rate (HDR) are based on the proper use of the MODCOD concept.

Especially, the Low-Density Parity Check (LDPC) code followed by the Amplitude-Phase Shift Keying (APSK) modulation will be the next-generation basis for HDR downlink [27] under the DVB-S2X standard [28].

In short, DVB-S2X is the second-generation extensions standard for Digital Video Broadcasting. This modulation-coding format for high data rate telemetry [29] is based on LDPC codes combined with eight selected modulation formats (QPSK, 8PSK, 8APSK, 16APSK, 32APSK, 64APSK, 128APSK, and 256APSK) and a wide range of code rates (1/4 to 9/10), ranging in spectral efficiency from 0.49 bit/s/Hz to 5.84 bit/s/Hz. As an example, the DVB-S2X 32APSK constellation is shown in Figure 3.

Thus, in the satellite context, the onboard computer receives the remote control to change the ACM mode parameters (MODCOD) from the Control and Tracking Centre through a continuous feedback link; the ground station reception system, ERG, performs real-time detection and monitoring of the energy per bit, *E_b_*, to interference, *I*_0_, plus noise, *N*_0_, spectral power densities, (*E_b_*/(*I*_0_ + *N*_0_)), information that induces the onboard transmitter itself to make the necessary adaptations.

The ACM technique can benefit cognitive satellite applications when combined with a CR method. Both concepts show how this new strategy can achieve better bandwidth and RF power efficiencies.

### 2.3. The Problem: FS Interferences on Low Orbit Satellite Downlinks

The X band is one of the frequency bands (8.025–8.400 GHz) regulated for satellite-to-ground data transmission applications in EESS [30]. Though, fixed service (FS) links that work within the same frequencies with beamwidth and maximum Effective Isotropic Radiated Power (EIRP) interfere at determined azimuth and elevation angles on the satellite receiving antenna. Figure 4 illustrates the interference scenario with the satellite receiver system.

The ITU Report [31] provides EESS downlink recommendations for system evaluation. The emission limit should not exceed the specified values, according to Table 1, and additional issues can be found in the Standard of CCSDS, 2021 [32].

This interference scenario occurs at INPE’s Cuiabá Earth station damaging CBERS4 and AMAZONIA1 satellite images of Amazon region. There, the EESS mission of the CBERS-4 (China-Brazil Earth Resources Satellite-4) satellite, launched in 2014, contains four cameras: PAN5/10, IRS, MUX, and WFI. The function of the wide field imager (WFI), with a resolution of 64 m and a coverage range (swath) of 866 km, is to monitor the Amazon rainforest. Figure 5 shows a frame (set of scenes) from the CBERS-4 (C-4) WFI image catalog covering twenty-six consecutive days. To examine how interference disrupts the WFI satellite images, we observe how the INPE image catalog is affected. In Figure 5a, we can see that the ideal framing of the image forms an almost perfect circular contour.

In Figure 5b, we can observe the harmful effects on the set of images with the lack of coverage points in certain locations in Brazil or South America, regions in northwest azimuths. These effects occur due to the existence of a FS link near INPE´s ERG in that geographic angular region.

### 2.4. Our Proposed Solution: A Combination of CR and MODCOD Techniques and Its Applications

Our proposed solution combines the CR concept in the underlay paradigm and MODCOD technique, in the ACM mode, to mitigate FS interferences on low-orbit satellite downlinks.

Calculation steps were developed in the cognitive paradigm for the operability of the exploitation technique. The following spectrum exploitation dimensions were considered: (i) power *P*; (ii) frequency *f_c_*; (iii) modulation MOD; (iv) coding COD; and (vi) bandwidth *BW*. The objective was to enable the satellite transmitter to improve the SINR (signal-to-interference plus noise ratio) at the receiver input.

Through link budget simulations for distinct standardized DVB-S2X—MODCOD configurations, we performed the troubleshooting strategy applied to spectral coexistence and data throughput maximization by adapting the MODCOD functionality. The variation of *m* bits per symbol in APSK modulations combined with appropriate code rates produced the bandwidth values as a function of both the symbol rate and the required SINR.

Considering that the specified BER is a function of SINR for each MODCOD adapted during each satellite pass over the ground station in the presence of interfering signals, the MODCOD selection depends on the interference-to-noise ratio (INR), which directly results from the measurement of *E_b_*/(*I*_0_ + *N*_0_) performed by the receiving system.

This adaptive technique is employed in the transmission channel of the WFI camera on the CBERS-4 and AMAZONIA-1 (AMZ-1) satellites, with a bit rate of 51.28 Mbps. Thus, the established MODCODs and their properties: designations, identifications (*ID*), spectral efficiencies (*η_tot_*), the relationship between *E_b_*/*N*_0_ and *E_S_*/*N*_0_, and the effective bandwidth of the modulated carrier are shown in Table 2. For example, based on Table 2, we can evaluate that for the MODCOD scheme QPSK 3/5 (*ID* = 8), the required *E_b_*/*N*_0_ for the specified BER (10^−7^) has a low value of *E_b_/N*_0_ = 1.480 dB, which increases the link margin and allows for operating with higher INR values.

On the other hand, for the MODCOD scheme 32APSK 8/9 (*ID* = 1), the required *E_b_/N*_0_ for the specified BER (10^−7^) has a high value of *E_b_/N*_0_ = 9.258 dB, the *INR* must be low to allow for this operating scheme.

In this way, by combining the results of link parameters with the MODCODs from Table 2, Table 3 presents the resulting MODCODs to be implemented in the ACM mode, considering the proper communication system of the CBERS-4 satellite’s WFI camera (EESS/FS coexistence). We maintained the specified link margin at 3 dB [33]. As a result, we obtain the maximum INR that each scheme supports.

In Figure 6, the results of Table 3 are graphically represented. We can see that as the interference power increases, the INR also increases, and the system should automatically select the appropriate MODCOD to support the current INR value while maintaining a constant bit rate transmission.

## 3. Results

Case studies—based on the previous results, this section quantifies the throughput for different assumptions. Firstly, we define the baseline case as nominal satellite operations, i.e., the maximum channel rate transmitted in the X band with fixed QPSK modulation without channel coding, as implemented on the CBERS-4 satellite.

### 3.1. Throughput for Fixed Modulation—Without Interference

In this case, the available bandwidth of 51.28 MHz for the WFI camera is limited to 51.28 Mbps. Note that the throughput (*T_hr_*) depends on the duration of the satellite pass in visibility with the ERG. By considering an average time of approximately 12 min [34], we have:

$T_{hr}=36.922\;Gbits$ (which represents the minimum throughput for WFI transmission).

### 3.2. Throughput for Fixed Modulation—With Interference

Implementing the same fixed QPSK modulation without channel coding in a hypothetical interference scenario, where the INR couples with the elevation angle of the receiving antenna, we obtain the following results presented in Table 4.

Note that the throughput 30.988 Gbits is below the minimum for successful image transmission (36.922 Gbits). In this case, we would observe failures in image processing.

### 3.3. Throughput Using MODCOD in Adapted ACM Mode

Subsequently, we analyze the effectiveness of our proposed strategy in Table 5, varying the ACM according to the INR in a satellite pass.

Thus, the throughput (82.878 Gbits) is higher than the minimum for WFI image transmission (computed as 36.922 Gbits). In this case, we would not expect failures in image processing.

### 3.4. Test Measurements at Cuiabá/Real Scenario

Expanding on the previous results, as a key contribution of the article, we apply our design strategy in an experimental scenario, considering field measurements recently conducted at the Cuiabá site in December 2022.

The ERG facility in the city of Cuiabá, Brazil, receives TM signals during the passage of the C-4 satellite, transmitting the TM image from the satellite’s payload cameras in real time.

We performed interference characterization by measuring the power of the interfering signal at the ERG receiving antenna as a function of the antenna azimuth and elevation angles and comparing this interference with the desired signal power levels received from the satellite. These results enable the acquisition of the carrier-to-interference ratio (*C/I*) and the interference-to-noise ratio (*INR*) necessary to implement the proposed strategy.

In our testing procedure, we operated with a high-performance ERG reception system with an 11.28 m diameter antenna. Figure 7 summarizes the implemented configuration. This reception system assumes a G/T performance of 35.5 dB.

Our solution, proposed in Section 2.4, can be applied based on the interference measured at the Cuiabá location. Figure 8 presents the power spectrum of the FS interference signal at the central frequency (*f_c_*) of 8308 MHz @ *BW* = 37 MHz. It is worth noting that the interference is located in the operational bandwidth of WFI of C-4, *f_c_* = 8290 MHz @ *BW* = 51.28 MHz. Furthermore, we can observe adjacent FS channels interfering in this spectrum band.

The harmful interference effect can be observed in the several black and color horizontal stripes on the moving-window image for a given satellite passage, presented in Figure 9. Each degraded WFI image line indicates 121k bits lost. The degree of harmful interference level can be checked when the image scenes are rejected by INPE’s quality assurance in the processing procedure (see lack of coverage points in Figure 5b).

The maximum tracked interference that generates these failures on the WFI images arises at *Az* = 333.5°. Further investigation on site identified the FS transmitting antenna at the corresponding azimuth, seen in Figure 10 below.

In this manner, the MODCODs are adapted during satellite visibility based on *INR* for antenna elevation angles (Table 6).

With current WFI link parameters for C-4, the *T_hr_* = 52.796 Gbits. As we can see, the proper operation of a low/medium complexity MODCOD mitigates the harmful effects by maintaining a higher margin than specified during satellite visibility, except for low elevation angles. The throughput for this configuration is about 1.4 times the minimum required of 36.922 Gbits. In this case, no processing failures would be expected for the images. We can recalculate the MODCOD by considering another approach to achieve the maximum throughput on the WFI channel. In return, we will have a high-complexity technology modulator.

This selection can be reassessed regularly, considering the reception systems of the ground stations. A low-cost station, for example, might need to use our case solution. On the other hand, when the reception system is composed of a high-performance receiver with high G/T, we can transmit a higher data rate, including other recorded data, as we have a greater throughput capacity.

### 3.5. End-to-End Communication System Emulation in Interfering Model

To emulate the previous scenario, Figure 11 shows the bench-emulated test setup of the TX and RX communication system with interference, with details of the RF devices/equipment used in the lab (Figure 12 is the setup picture). The specifications and description of the transmission and reception communication system are detailed below.

In Figure 11, we have the CORTEX equipment, indicated as (1)—high-data rate receiver Cortex Series from the French company Zodiac Safran, which has a dual function in this setup. Firstly, this equipment is a TEST MODULATOR that generates the modulated carrier signal with the WFI camera information data provided by the WFI CAMERA SIMULATOR-DEMO 1 (2). In this sense, we emulated the satellite data transmitter in the communication system. For practicality, the bit rate *R_b_* was programmed at 50 Mbps (close to the actual WFI value of 51.28 Mbps).

The TEST MODULATOR output is combined with a signal generated by the Rohde Schwarz SMJ 100A synthesizer (3), which emulates the interfering signal. The actual FS signal specification was used to set the synthesized signal, with *BW* = 37 MHz and QPSK modulation. The interference output power is varied according to the required SINR to measure the interference effects on the received images. The various microwave devices (attenuators, isolators, and couplers) are used for impedance matching and adjusting the signal level properly.

At the output of the combiner (4), we have the compound signal: desired and interference signal plus additive white Gaussian noise (AWGN), which is looped back to CORTEX equipment (1). The demapping of the I/Q signal from the MAPSK modulation and the decoding are performed in the DEMODULATOR on CORTEX. 

Finally, the I/Q bit stream at the DEMODULATOR output, corresponding to the camera image data received by ERG, is sent to the acquisition board in the computer DEMO 2 (5) for image processing, according to Figure 13.

The total losses between ‘IF TEST’ (MODULATOR OUT) and ‘IF NOMINAL’ (DEMODULATOR IN) is *L_T_* = 19.0 dB.

#### 3.5.1. Test 1 Results (No Interference)

Signals configuration at HDR CORTEX (‘IF TEST’): *C* = −9.6 dBm; *N*_0_ = −102 dBm/Hz. Then, ‘IF NOMINAL’ gives: *C* = −28.6 dBm and *N*_0_ = −121 dBm/Hz.
(1)$$∴\frac{C}{N_{0}}=-28.6+121=92.4\;dBmHz.$$

Since *R_b_* = 50 Mbps (WFI–AMZ-1), and
(2)$$\frac{E_{b}}{N_{0}}=92.4-10\log_{10}\left(R_{b}\right),∴\frac{E_{b}}{N_{0}}=15.4\;dB,\;$$

*E_b_/N*_0_ = 15.4 dB is equal to the link budget parameter of the AMZ-1 satellite WFI camera.

Cortex measurements:

*N*_0(CRTX)_ = −99.7 dBm/Hz − 19 dB = −118.7 dBm/Hz, and *C* = −29 dBm. Since,
(3)$$\frac{E_{b}}{N_{0}}=89.7-10\log_{10}(R_{B}),∴\frac{E_{b}}{N_{0}}=12.7\;dB\;.$$

Spectrum analyzer measurements (with interference):

*I_(SPECT)_* = −36.8 dBm and *N*_0_ = −118.3 dBm/Hz,
$$⟹I_{0}=-36.8-77=-113.8\;dBm/Hz.\;\mathrm{Since},$$
(4)$$\frac{I_{0}}{N_{0}}=-113.8+118.3,\;∴\frac{I_{0}}{N_{0}}=4.5\;dB=INR\;,$$
and,
(5)$$\frac{E_{b}}{N_{0}+I_{0}}=\frac{SNR}{1+INR},∴\frac{E_{b}}{N_{0}+I_{0}}=12.7-10\log_{10}3.88\approx 6.9\;dB\;.$$

Measuring the relation *E_b_*/(*N*_0_ + *I*_0_) on CORTEX, we have:(6)$$\frac{E_{b}}{N_{0}+I_{0}}=7\;dB.$$

Measure (6) is close to that calculated in (5). So,
(7)$$1+INR=10^{\frac{12.7-7}{10}},∴INR\approx 4.3\;dB.$$

Comparing (7) with the result in (4), we can conclude that the INR values calculated by the CORTEX and spectrum analyzer measurements have extremely low variation, validating our test setup.

#### 3.5.2. Test 2 Results (Interference Increasing)

To re-evaluate the analyses, we increase the interfering signal level:$$I_{\left(SPECT\right)}=-36.8\;dBm@P_{T_{\left(SMJ\right)}}=-11.0\;dBm.$$

Tuning the SMJ generator to give:$$\frac{E_{b}}{N_{0}+I_{0}}=10.5\;dB,$$
we have:$$P_{T_{\left(SMJ\right)}}=-17.5\;dBm\left(checked\right)\;.$$
(8)$$INR\prime =INR-\left(-11.0+17.5\right)dB,∴INR\prime =4.5-6.5=-2\;dB.$$

Since,
(9)$$1+INR\prime =10^{\frac{12.7-10.5}{10}}⟹INR\prime =1.65∴\;INR\prime \approx -1.9\;dB.$$

The value (9) slightly differs from that previously calculated in (8). Based on the consistency of our communication-interfering system test results, we can extend the computations on the system with MODCOD schemes, which aim to mitigate interference levels, optimizing the required *E_b_/N*_0_ according to the variation of INR.

#### 3.5.3. Test 3—MODCOD Results

As an example, we calculate the maximum INR for the MODCOD QPSK 3/5 (ID = 8) @ *η_tot_* = 1.883. From (2), the received SNR at the demodulator input is 15.4 dB, and the minimum specified is:$${\left(\frac{E_{b}}{N_{0}}\right)}_{required}=1.5\;dB,∴M=13.9\;dB.$$

For *M* = 3 dB, we have:(10)$$\frac{E_{b}}{N_{0}+I_{0}}=4.47\;dB⟹\frac{C}{I}=5.4\;dB.$$

Like,
(11)$$∆=1+INR\leq M⟹1+INR\leq 10^{\frac{13.9}{10}}∴INR\leq 13.7\;dB.$$

For downlink practical checking, the earth station antenna gain, *G_ant_*(RX) = 55 dB (from budget parameters):$$C=EIRP-losses+G_{{ant}_{RX}}=14.5-182.14+55∴C=-112.64\;dBm.$$

Like,
$${\left(\frac{C}{N_{0}}\right)}_{donwl}=95.5\;dBmHz,∴N_{0}=C-95.5=-208.1\;dBm/Hz,$$
$$⟹\frac{I_{0}}{N_{0}}=13.7,∴I_{0}=-194.4\frac{dBm}{Hz}.$$

Given:$$I=I_{0}+10\log_{10}\left(R_{b}\right)-10\log_{10}\eta_{tot}=-194.4+77-0.75=-118.15\;dBm,$$
(12)$$∴\frac{C}{I}=5.5\;dB\;.$$

Results (10) and (12) demonstrate the consistency. For the remaining MODCODs, the calculation sequence follows the same steps.

## 4. Discussion

Regarding the on-site results presented in Section 3.4, we considered the link closure with a minimum margin of 3 dB, which allows a lower technical complexity modulator system. On the other hand, if we considered an approach to maximize throughput, where we could transmit a larger volume of data, including playback recorded passages, we would have to adapt to higher-order MODCODs demanding a modulator of higher technical complexity for this development. This trade-off must be re-evaluated when considering satellite reception systems. Therefore, the results show the minimum MODCOD adaptation to support the INR as a function of antenna elevation for the azimuth point of highest interference.

Furthermore, as shown in Figure 8, the on-site measurements indicate that the interference behavior around the receiving antennas does not have theoretically predicted distributions. These variations are due to the reflections around the ERG and the different front–back ratios of the antennas. Additionally, the signals can be received by the secondary lobes of the antennas.

In the emulated scenario of Section 3.5, the end-to-end communication system allows the variation of SINR for constantly monitoring the image scenes processed from the WFI camera simulator. Consequently, for the moving window of WFI images without failures shown in Figure 13, we set a link margin of *M* = 3 dB for the specified BER = 10^−7^ in the link budget. In other words, considering QPSK modulation and no error correction code, the SINR relation is given by:(13)$$\frac{E_{b}}{I_{0}+N_{0}}=\frac{E_{b}}{N_{0}}\left(required\right)+M=11.3+3=14.3\;dB.$$

As a secondary measurement step, in order to find the threshold level, we increase the interference until we have the no-failure threshold, as shown in (14) and demonstrated in Figure 14.
(14)$$\frac{E_{b}}{I_{0}+N_{0}}=8.2\;dB⟺BER≅1.2\times 10^{-4}\;.$$

To illustrate the failures in the image processing, we reduce the SINR to 5.2 dB, and we have the moving window depicted in Figure 15, which shows the processing failures in the image. The black stripes in the image highlight these failures.

## 5. Conclusions

This article proposed a technical solution that allows spectral coexistence between terrestrial fixed links and the downlink of Earth exploration LEO satellites. The solution maximizes throughput in interfering conditions while maintaining the mission BER requirement within the specified value for the interference plus noise levels during the satellite pass transmitting the image data telemetry. To achieve the target, we developed a strategy that combined the underlying CR exploitation with the adaptive MODCOD technique. The method adjusts the MODCOD to receive error-free data images based on the SINR per antenna angle. There is a numeric evaluation of the quality of the images given by the specified BER of 10^−7^. In our emulation, we analyze qualitatively the impact of the drop in signal-to-noise ratio on image processing.

## Figures and Tables

**Figure 1 sensors-23-07234-f001:**
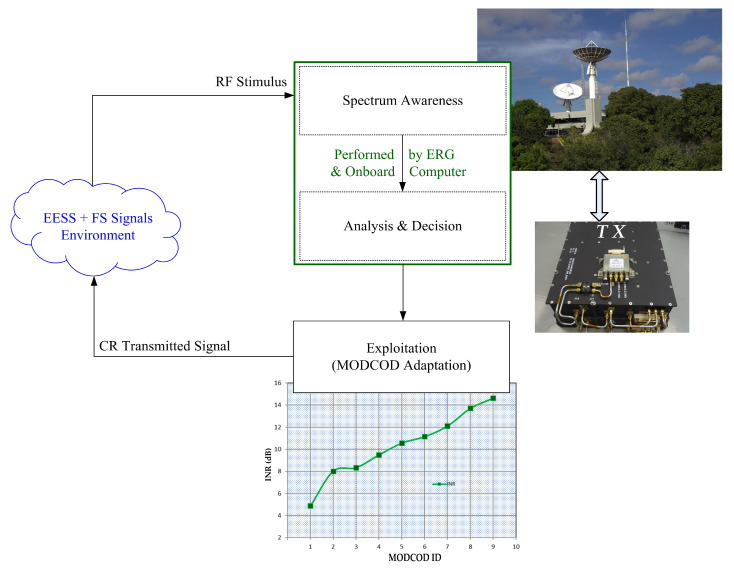
Cognitive cycle for EESS-FS coexisting system.

**Figure 2 sensors-23-07234-f002:**
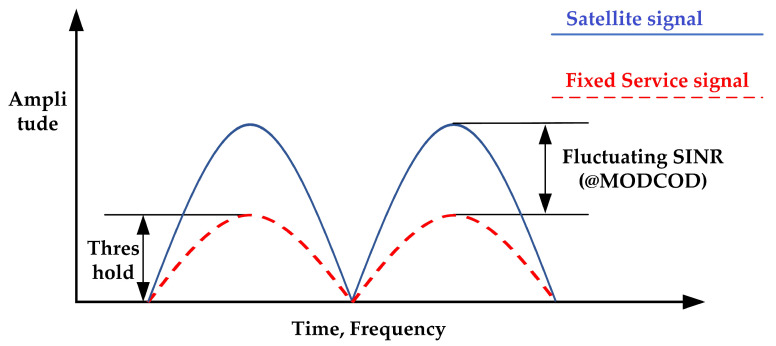
Representation of underlay paradigm. Source: adapted from [11].

**Figure 3 sensors-23-07234-f003:**
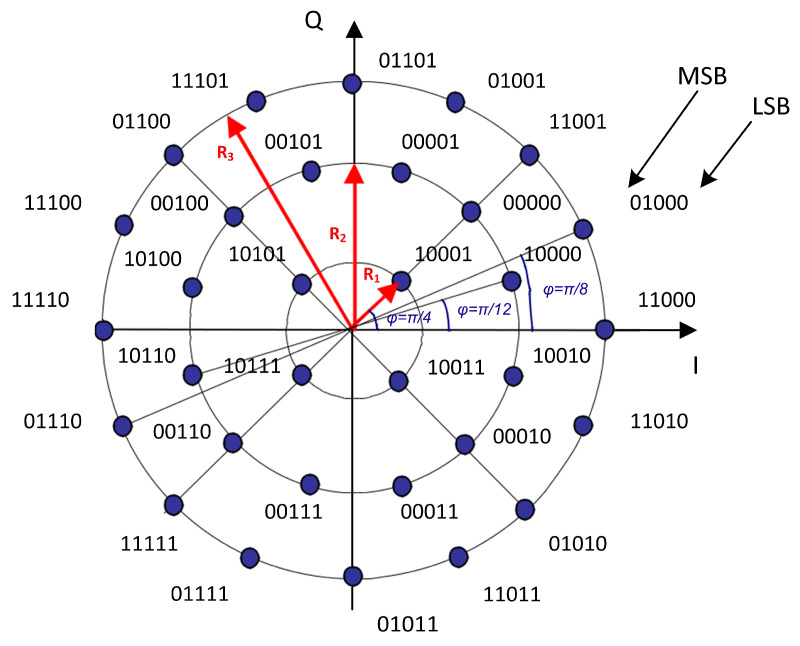
Bit-mapping constellation for the 32APSK modulation.

**Figure 4 sensors-23-07234-f004:**
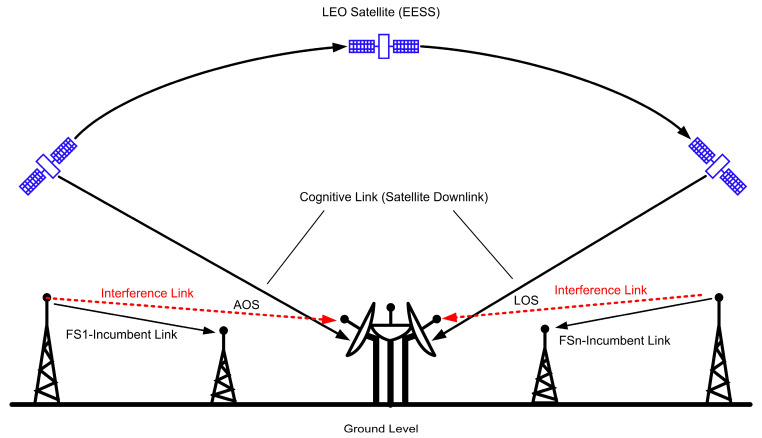
LEO satellite downlink scenario.

**Figure 5 sensors-23-07234-f005:**
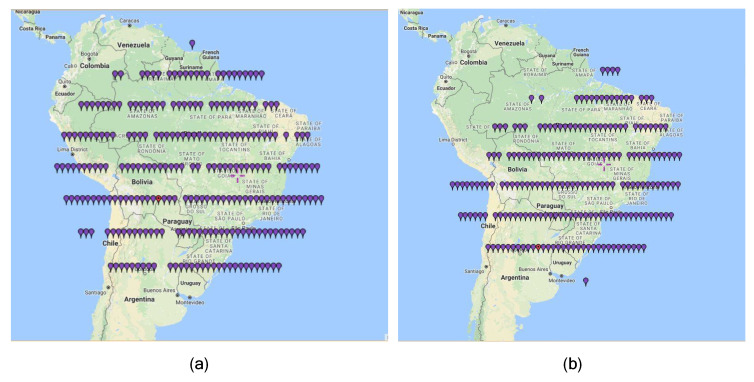
Depiction of images from C-4 WFI (26 days INPE’s catalog). Source: adapted from DGI-INPE (2018): (**a**) complete set; (**b**) set with failures.

**Figure 6 sensors-23-07234-f006:**
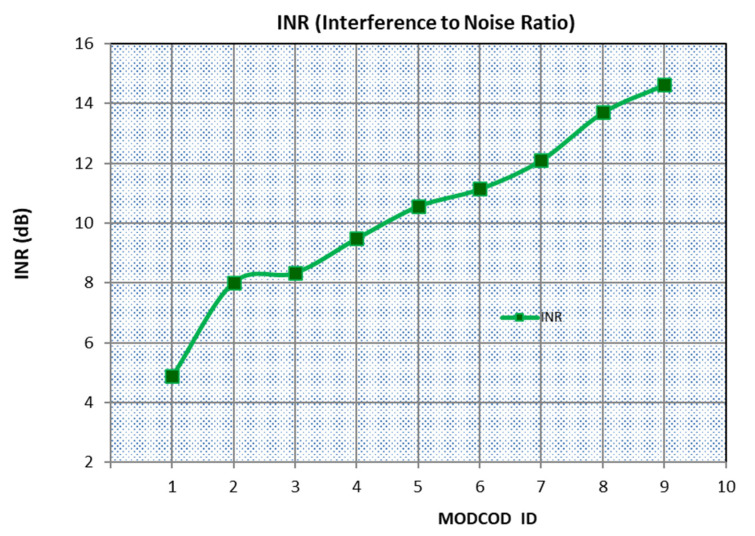
Curve of *INR* variation versus applied MODCOD.

**Figure 7 sensors-23-07234-f007:**
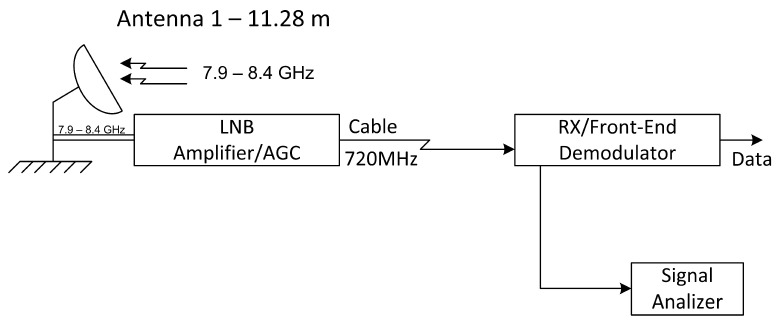
Setup in the ERG system.

**Figure 8 sensors-23-07234-f008:**
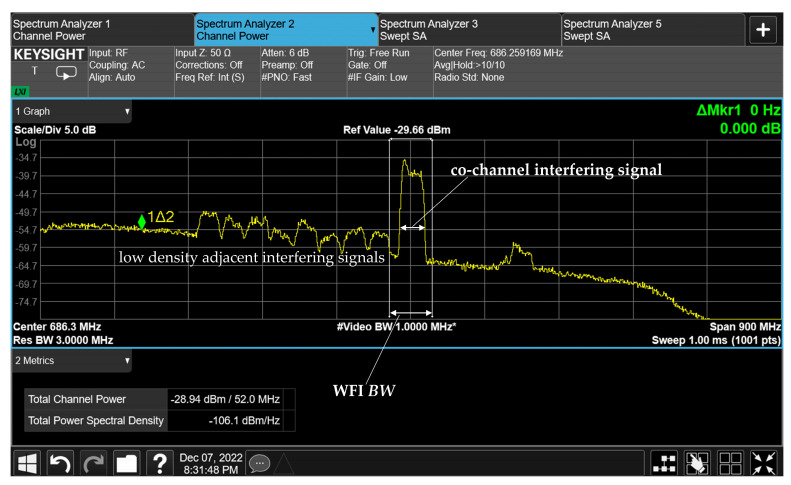
Power spectrum of the interference, *f_c_* = 8308 MHz.

**Figure 9 sensors-23-07234-f009:**
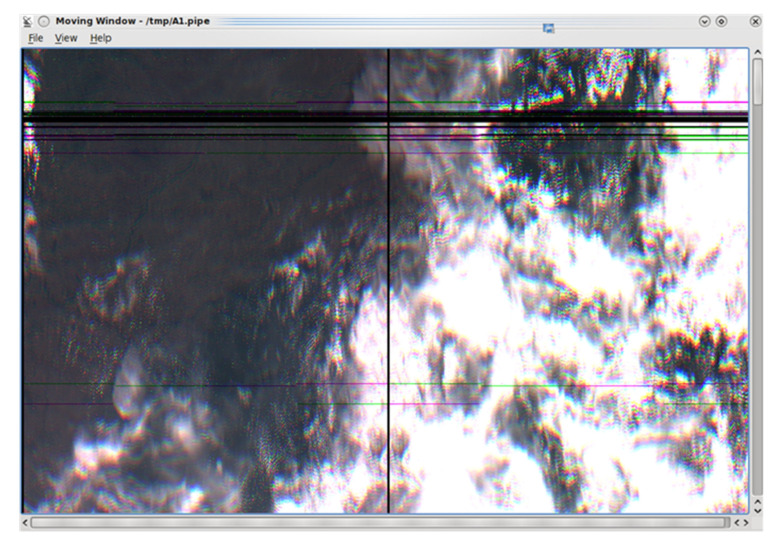
WFI images with harmful interference effects.

**Figure 10 sensors-23-07234-f010:**
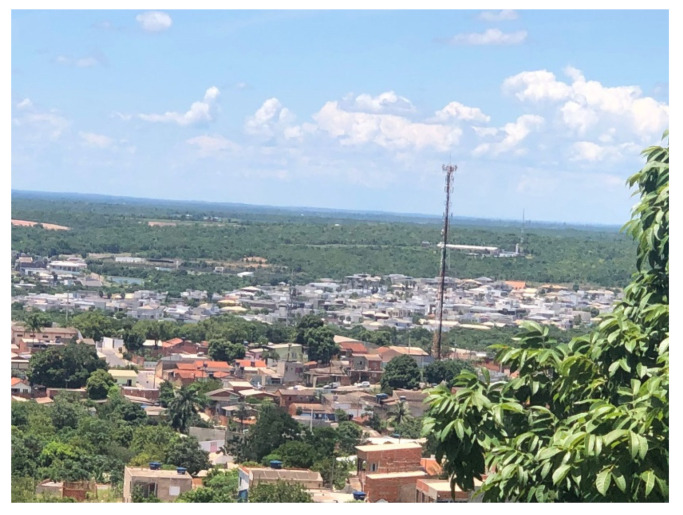
Interfering fixed station.

**Figure 11 sensors-23-07234-f011:**
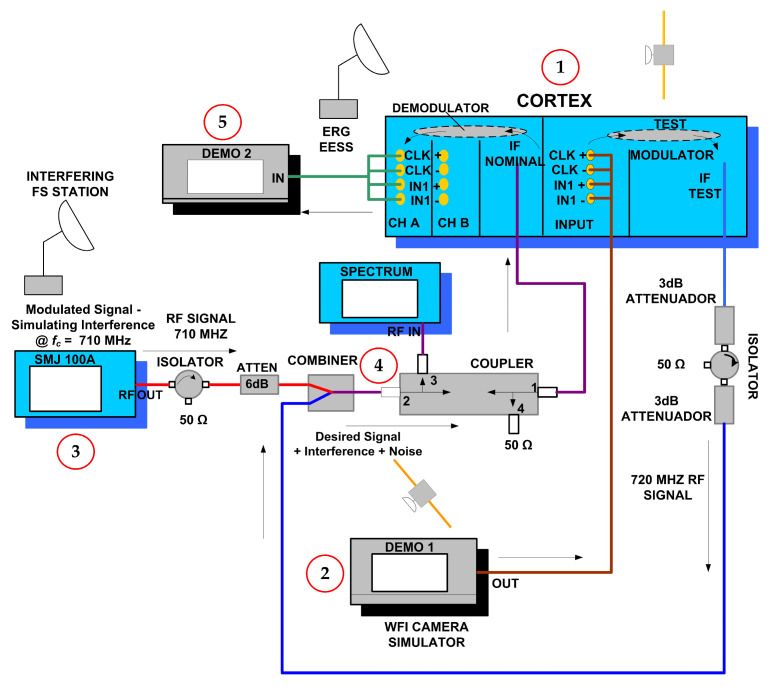
Setup of the communication system.

**Figure 12 sensors-23-07234-f012:**
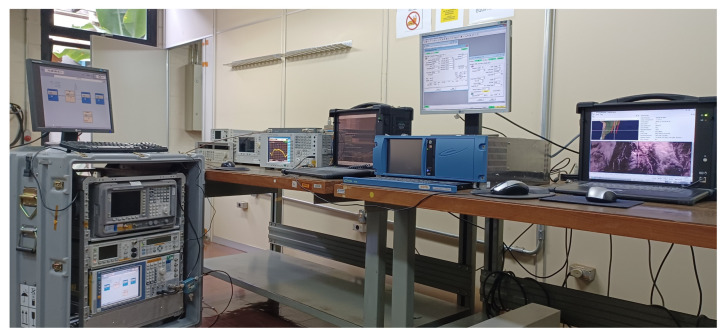
Picture of the communication system setup.

**Figure 13 sensors-23-07234-f013:**
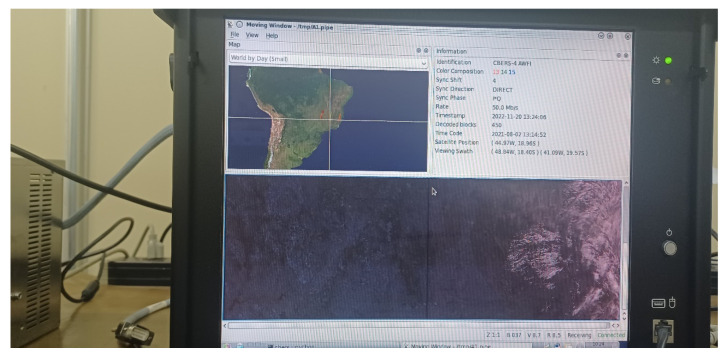
Moving window of WFI images with no failures.

**Figure 14 sensors-23-07234-f014:**
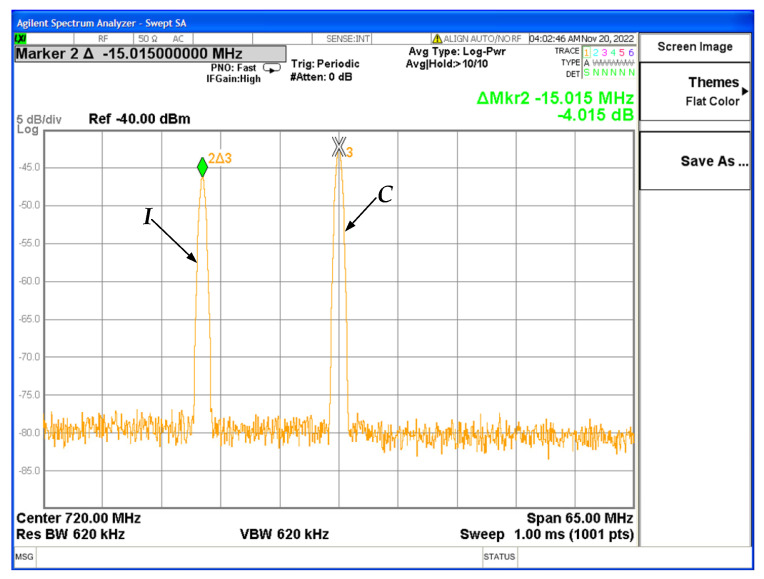
Power spectrum of the unmodulated WFI signal (*C*) and the interference (*I*).

**Figure 15 sensors-23-07234-f015:**
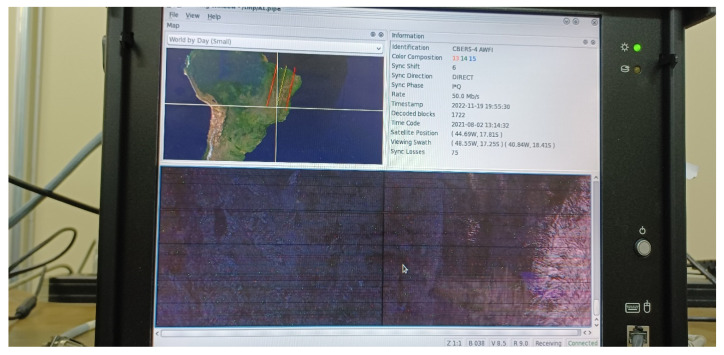
WFI moving window with processing failures.

**Table 1 sensors-23-07234-t001:** EESS emission limit recommended by ITU.

Frequency Band	Service	Limit in dB (W/m^2^) for Angle of Arrival (δ) above the Horizontal Plane			Reference Bandwidth
8025–8500 MHz	Earth Exploration Satellite	0–5°	5–25°	25–90°	4 kHz
		−150	−150 + 0.5(δ − 5)	−140	

**Table 2 sensors-23-07234-t002:** Adaptive MODCOD in DVB-S2X.

MODCOD	*ID*	*η_tot_*	*E_S_/N*_0_ (dB) Ideal Frame of 64,800 Bits	*E_b_/N*_0_ [dB]	Symbol BW (MHz)
QPSK 1/3	9	0.656448	−1.24	0.587996705	78.12
QPSK 3/5	8	1.188304	2.23	1.480724410	43.15
QPSK 5/6	7	1.654663	5.18	2.992904443	30.99
QPSK 9/10	6	1.788612	6.42	3.894838598	28.67
8PSK 3/4	5	2.228124	7.91	4.430606434	23.01
8PSK 5/6	4	2.478562	9.35	5.408002130	20.69
8PSK 9/10	3	2.679207	10.98	6.699937308	19.14
16APSK 5/6	2	3.300184	11.61	6.424618456	15.54
32APSK 8/9	1	4.397854	15.69	9.257591924	11.66

Note: given the system spectral efficiency *η_tot_* = the ratio between the energy per information bit and single-sided noise power spectral density: *E_b_*/*N*_0_ = *E_s_*/*N*_0_ − 10log_10_(*η_tot_*).

**Table 3 sensors-23-07234-t003:** Results of the link budget parameters from the implemented MODCOD.

Overall Link Budget	$QPSK\frac{1}{3}$ *ID* = 9	$QPSK\frac{3}{5}$ *ID* = 8	$QPSK\frac{5}{6}$ *ID* = 7	$QPSK\frac{9}{10}$ *ID* = 6	$8PSK\frac{3}{4}$ *ID* = 5	$8PSK\frac{5}{6}$ *ID* = 4	$16APSK\frac{5}{6}$ *ID* = 2
Received *C/N*_0_ [dBHz]	95.5						
Received *E_b_/N*_0_—loss [dB]	15.4						
Implement. Loss [dB]	3.0						
Demodul. Loss [dB]	3.0						
*E_b_/N*_0_ @ BER = 1 × 10^−6^ [dB]	0.6	1.5	3.0	3.7	4.4	5.4	6.4
*C/I* [dB]	1.9	5.4	8.4	9.6	11.3	12.8	15.2
*E_b_/I*_0_ [dB]	3.9	4.9	6.5	7.6	8.2	9.4	10.7
Received *E_b_*/(*N*_0_ + *I*_0_) [dB]	3.58	4.48	5.98	6.91	7.45	8.42	9.43
Spec Margin (*M*) [dB]	2.99	3.00	2.99	3.01	3.02	3.02	3.001
*INR_max_* = *I*_0_*/N*_0_ [dB]	14.62	13.69	12.10	11.14	10.56	9.49	8.34

**Table 4 sensors-23-07234-t004:** Throughput for no-coding QPSK/interference scenario (*INR* vs. elevation).

Elevation (°)	INR (dB)	% of Time ^(2)^ (CCSDS, 2013) [15]	*R_b_* ^(1)^ Mbps	*T_hr_* ^(2)^ Gbits
90	8.01	16.6	51.28	30.988
75	8.37	13.6	51.28	
60	9.44	25.2	51.28	
30	11.45	35.4	39.65	
5.7	13.70	9.2	6.45	

Notes: ^(1)^ the bandwidth is limited to 51.28 MHz; ^(2)^ visibility of 12 min (elevation of 0° a 180°).

**Table 5 sensors-23-07234-t005:** Throughput X MODCOD (in ACM).

Elev. (°)	*INR* dB	MODCOD	ID	% of Time ^(2)^	*R_b_* ^(1)^ Mbps	*T_hr_* ^(2)^ Gbits
90	8.01	8PSK 9/10	2	16.6	137.39	82.878
75	8.37	16APSK 5/6	3	13.6	169.23	
60	9.44	8PSK 5/6	4	25.2	127.1	
30	11.45	QPSK 9/10	6	35.4	91.92	
5.7	13.70	QPSK 3/5	8	9.2	51.28	

Notes: ^(1)^ the bandwidth is limited to 51.28 MHz; ^(2)^ visibility of 12 min (elevation of 0° a 180°).

**Table 6 sensors-23-07234-t006:** Calculation of throughput X adaptive MODCOD—C-4 passage.

Elev. (°)	*I* (dBm)	*C* (dBm)	*C/I* (dB)	INR (dB) ^(1)^	MODCOD (ID)	*R_b_* (Mbps)	Time ^(2)^ %
10	−21	−20.2	0.8	18.97	9⇒ ⇒M = −1.05 dB	0	5.88
20	−32.5	−19.7	12.8	6.97	8	60.94	5.88
30	−40.5	−18.7	21.8	−2.03	7	80.85	5.88
40	−47.5	−18.4	29.1	−9.33	6	91.72	5.88
50	−40.5	−17.9	22.6	−2.83	7	80.85	5.88
60	−45	−16.4	28.6	−8.83	6	91.72	5.88
70	−44	−16.2	27.8	−8.03	7	80.85	5.88
77.8	−26.5	−14.7	11.8	−7.97	7	80.85	5.88
90	−44	−14.1	29.9	−10.13	6	91.72	5.88
100–140				−2.53–−8.03	7	80.85	29.4
150–170				0.97–5.47	8	60.94	17.64

^(1)^ *BW* = 52 MHz/*BWI* = 37 MHz; ^(2)^ uniform distribution between 10° and 170°. (Unlike previous analyzes where we considered a distribution specified by CCSDS, in this case we assumed a uniform distribution for our real scenario for clarity of presentation).

## Data Availability

The data presented in this study are available on request from the corresponding author.

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
