# Peer review of "Cognitive Radio Strategy Combined with MODCOD Technique to Mitigate Interference on Low-Orbit Satellite Downlinks"

_sensors, 2023, doi:10.3390/s23167234_

Round 1
Reviewer 1 Report
The work focuses on mitigating interference on low-orbit satellite downlinks using Cognitive Radio (CR) and Adaptive MODulation and CODing (MODCOD) techniques. The work is well presented and the theme is interesting. However, revisions are required for further processing of the paper.
1. The problem being addressed in this work is interesting but it is not adequately motivated. The literature search is impoverished. Authors are required to discuss quality literature to identify the gaps in this domain to better motivate the problem being addressed.
2. Figures 1 and 2 show generic information that we already know. It is not clear what the authors want to achieve by adding these figures to the paper. We expect to see the actual system model being studied in the current work.
3. Figure 5 is not readable. The texts are blurred. Only legible figures are accepted in scientific papers for this journal.
4. In Figure 9, it is not clear what constitutes harmful interference, and to what degree interference can be termed harmful and not harmful.
5. In Figure 10, how do you address the problem of interference from co and adjacent channels?
6. It appears that the results are not described and discussed sufficiently. Please, elaborate on the results and state any assumptions made in the design and experimentation.
7. I suggest that you add and discuss a few more related works from 2023.
A minor English check is required.
Reviewer 2 Report
Authors investigated the adaptive modulation and demodulation technique to increase RF power and spectral efficiencies in CR environment. However, before acceptance, major corrections are required:
1. Abstract should be specific. Not need numbering.
2. Introduction should be rewrite. Too many para.
3. Related works NOT discuss. Please add related work in Introduction or another subsection.
4. Please write down the contribution end of the Introduction.
5. How to get Table 2 and Table 3 data, it is from experiment or theoretical. NOT clear.
6. Theoretical analysis should be in front of the result and discussion (3.5.1. Test 1 Results (No Interference)).
7. System model design is NOT clear. Please improve.
8. Results section is NOT clear. Please improve.
9. It is recommended to check the full paper by native English speaker.
It is recommended to check the full paper by native English speaker.
Round 2
Reviewer 1 Report
The authors have addressed my earlier comments
Minor English check is required.
Reviewer 2 Report
Thanks for correction.
Not required.